# Creating a Japanese version of the attitudes toward dreams scale: Attitude toward dreams may predict sleep disorders

**Shinya Okuyama**[1]*, **Kenta Nozoe**[1,2], **Kazuhiko Fukuda**[1,2], **Takamasa Kogure**[3], **Shoichi Asaoka**[1,2]

1 Sleep Research Institute, Edogawa University, Nagareyama-shi, Chiba, Japan, 2 College of Sociology, Edogawa University, Nagareyama-shi, Chiba, Japan, 3 Paramount Bed Sleep Research Laboratory, Paramount Bed Colorado., LTD., Koto-ku, Tokyo, Japan

* okuyamas@edogawa-u.ac.jp

## Abstract

Previous studies have identified two factors that influence people's attitudes toward dreams: positive and negative. However, cultural differences may influence these attitudes. This study aimed to develop a Japanese version of a scale to assess attitudes toward dreams and to examine factors related to Japanese participants' attitudes toward dreams. These factors included dream attitudes, dream-recall frequency, and various sleep-related variables (Pittsburgh Sleep Quality Index, sleep apnea, restless legs syndrome, insomnia, and rapid eye movement sleep behavior disorder), personality traits, and quality of life. Additionally, we aimed to develop a new scale to measure dream attitudes. Nationwide data were collected from eight regions in Japan. We included 1,478 participants (728 men and 750 women) and assessed their dream attitudes, dream-recall frequency, several sleep variables, personality, and quality of life. We confirmed a two-factor structure similar to that previously described: Factor 1, named "meaning of dreams," reflects individuals' attitude to interpreting dreams, while Factor 2, named "no meaning of dreams," represents the attitude that dreams have no meaning. Both factors were significantly correlated with several variables. Factor 1 was more strongly correlated with rapid eye movement sleep-related variables. Scores for both factors declined with age, especially among individuals in their 60s and 70s. Factor 1 and dream-recall frequency showed significant sex differences among individuals in their 20s and 30s. Overall, we identified a two-factor structure of attitudes toward dreams, aligning with the Schredl scale. Factor 1 (meaning of dreams) was positively correlated with several dream- and rapid eye movement-related variables. Notably, a higher Factor 1 score in older individuals may indicate an increased risk of rapid eye movement sleep behavior disorder. The observed sex differences in dream-recall frequency may be based on differing attitudes toward dreams, particularly the tendency to find meaning in them, especially among younger individuals.

**Data availability statement:** "All relevant data are within the paper and its Supporting Information files."

**Funding:** This research was funded by PARAMOUNT BED CO., LTD. Dr. Kogure is affiliated with the Paramount Bed Sleep Research Institute. Dr. Kogure was also partially involved in the design of this study and the revision of the manuscript. However, this study is not directly related to the interests of PARAMOUNT BED CO. LTD. The purpose of this study is purely scientifically motivated, and the results obtained are not necessarily directly related to the interests of PARAMOUNT BED CO LTD. This does not alter our adherence to PLOS ONE policies on sharing data and materials.

**Competing interests:** Dr. Takamasa Kogure belongs to PARAMOUNT BED CO., LTD., however, their interests are not competing with the purpose of this study. This does not alter our adherence to PLOS ONE policies on sharing data and materials.

## Introduction

It is well established that there is a substantial sex difference in dream-recall frequency, with women typically recalling dreams more often than men do [1]. Schredl [2] identified significant effects of nocturnal awakenings and interest in dreams on dream recall, noting that women wake up more frequently during the night and have a more positive attitude toward dreams than do men. Schredl has conducted multiple studies on attitudes toward dreams [3–6], revealing a two-factor structure: positive and negative attitudes toward dreams [3]. Although these factors may seem to represent opposing ends of a single continuum, Schredl et al. argued that they are associated with distinct variables, suggesting they are not simply inverses of one another. Their confirmatory analysis supported this two-factor structure. Factor 1 (positive) reflects the attitude of actively seeking meaning in dreams, while Factor 2 represents a lack of recognition of significant meanings in dreams [6]. Other studies have similarly found that some individuals place high value on the relevance of dream content to real-life events [7,8].

While the two-factor structure has been consistently identified in several studies [3–6,9,10], Kodama [11] identified three factors related to attitudes toward dreams among Japanese participants: Factor 1 (meaning and utilization of dreams), Factor 2 (value of dreams), and Factor 3 (influence of dreams). This suggests that attitudes toward dreams may vary across cultures, leading to different factor structures. Therefore, this study aimed to develop a Japanese version of a scale to assess attitudes toward dreams and to examine the factors associated with Japanese participants' attitudes toward dreams.

## Materials and methods

Previous research has shown that the frequency of dream recall decreases with age [12,13]. Therefore, participants in this study were selected to range in age from their 20s–70s. Additionally, we examined the impact of other factors such as sleep habits, personality traits, and metacognitive traits (i.e., traits related to knowledge and awareness of one's memory) on dream-recall frequency and the relationships among these factors. Furthermore, the relationship between attitudes toward dreams and mental health was explored to address potential negative health effects associated with dream-related attitudes.

### Participants

An online research company (Cross Marketing Inc., Tokyo, Japan) conducted the survey. The recruitment period for participation in this study was from March 18 to March 22, 2022. Initially, 1,680 participants were selected for the survey. All participants were adults aged 20–70 years, residing in Japan and registered with Cross Marketing Inc. Some participants had previously participated in other surveys; however, the data used in this study were collected specifically for this project during the designated recruitment period, and no data from previous surveys were included. To assess sleep-related parameters, we used the Japanese version of the Pittsburgh Sleep Quality Index (PSQI-J). While Cross Marketing Inc. ensured that all questionnaire

items were completed (i.e., there were no missing responses), the data from 202 participants were excluded from analysis because of internally inconsistent responses regarding bedtime, wake-up time, and sleep duration, which made it impossible to accurately calculate their global PSQI scores. Responses were considered inconsistent if they included, for example: (1) a reported sleep latency of 120 min, a sleep duration of 8 h, and a total time in bed of only 7 h (i.e., logically impossible); (2) a bedtime of 10:00 PM and a wake-up time of 8:00 AM, but a sleep duration of only 2 h without any reported nighttime awakenings; or (3) a sleep duration that exceeded the total time in bed (e.g., 8.5 h of sleep during 6 h in bed). These types of discrepancies, likely due to input errors or misunderstandings of the questionnaire, prevented the accurate calculation of key PSQI components, such as sleep efficiency. As a result, the final analytical sample consisted of 1,478 participants (728 men, 750 women) who provided logically consistent and complete data on the PSQI-J.

Nationwide data were obtained (from approximately 0.001–0.002% of the total population) from eight regions of Japan: Hokkaido, Tohoku, Kanto, Chubu, Kinki, Chugoku, Shikoku, and Kyushu-Okinawa. For this study, we analyzed data from 1,478 individuals (728 men and 750 women) who provided consistent responses about sleep-related factors such as bedtime, wake-up time, and sleep duration in the PSQI-J, a screening tool for sleep disorders.

Before the study began, all participants were informed that their responses would be analyzed only as group data, with no personal information disclosed, and that the results would be used for academic research and other purposes, including lectures and secondary use. The study was conducted in accordance with the principles of the Declaration of Helsinki, and written informed consent was obtained from all participants.

The Ethics Review Committee of Edogawa University approved the research protocol (approval No. R03-026A).

## Survey items

The survey items used in this study included the following: participant characteristics; dream-recall frequency; nighttime awakenings; rapid eye movement (REM) sleep behavior disorder (RBD) [14]; questionnaire for the evaluation of obstructive sleep apnea (STOP-Bang) [15] (only the four STOP items: snoring, tiredness, observed apnea, and high blood pressure were included; the BANG items—BMI, age, neck circumference, and gender— were not included and were therefore not obtained from the participants); Restless Legs Syndrome scale (RLS scale: assessing the frequency of RLS symptoms) [16]; Athens Insomnia Scale [17]; PSQI-J [18]; Short Form-8 (SF-8) for Health-Related Quality-of-Life Assessment [19]; NEO Five-Factor Inventory (NEO-FFI) for personality evaluation [20]; Japanese version of the Meta-memory in Adulthood Questionnaire [21]; Dream Propensity Scale [22]; Munich Chronotype Questionnaire for biological rhythm assessment [23]; nap habits; Japanese version of the Nightmare Distress Questionnaire [24]; and attitudes toward dreams. Although the above items were included in the survey, data from the Munich Chronotype Questionnaire, nap habits, and the Japanese version of the Nightmare Distress Questionnaire were not converted into numerical scores for statistical analysis. Therefore, these variables were excluded from the statistical analyses in the present study.

## Questionnaire scoring and cutoff criteria

In this study, we used a variety of psychological and sleep-related questionnaires, each with specific scoring characteristics and interpretive guidelines. The Japanese version of the RBD scale (RBDQ-J) serves as a screening tool for REM sleep behavior disorder, with scores of 5 or higher suggesting probable RBD. For the STOP-Bang questionnaire, only the four STOP items (snoring, tiredness, observed apnea, and high blood pressure) were included in this study; therefore, no established cutoff score is applicable. The RLS scale is used to assess symptom frequency, with higher scores indicating more frequent symptoms. The Athens Insomnia Scale is used to screen for insomnia, with scores of 6 or higher suggesting possible insomnia. The PSQI-J is used to measure subjective sleep quality, with a global score of 6 or higher indicating poor sleep quality. The Japanese version of the Metamemory in Adulthood Questionnaire is used to evaluate beliefs and awareness regarding memory functioning, and the NEO-FFI is used to assess five personality traits. In both instruments, higher scores reflect stronger trait expression. The Dream Propensity Scale captures dream-related tendencies;

however, no defined cutoff score is provided. The SF-8 is used to measure health-related quality of life using norm-based scoring (mean = 50, standard deviation = 10), with higher scores indicating better perceived health status.

## Data collection

An online survey was used to gather data on the aforementioned questionnaire items. In addition, questions about attitudes toward dreams were developed based on four previous studies [2,4,9,11], and participants' responses were collected. The details of the questions posed to participants are explained in the following paragraph.

## Attitudes toward dreams

Forty-two questions were developed to assess attitudes toward dreams. These included 8 questions from the Mannheim Dream Questionnaire [9], 10 from the Attitudes Toward Dreams Scale [2], 6 from a previous study on dream attitude [4], and 18 from the Dream Attitudes Scale [11]. Since these studies used either a five-point scale [2,4,9] or a four-point scale [11], the questionnaire in this study was standardized to a seven-point scale. Responses were scored on a seven-point scale ranging from "not at all applicable" to "very applicable." Weng [25] reported that "whether language labels are provided for all categories or only for the endpoints, having fewer response categories results in lower reliability." Additionally, Krosnick and Presser [26] provided comprehensive guidelines for survey and scale design, noting that a seven-point scale "can more accurately reflect respondents' opinions by including a middle option." However, neither of these studies concluded that the seven-point scale is superior to other methods. Nevertheless, in this study, a seven-point scale was used in the survey on attitudes toward dreams to enhance the reliability of the results.

## Analysis

The results of the Attitudes Toward Dreams Scale questionnaire were first subjected to exploratory factor analysis (EFA) to extract the factor structure. Items with factor loadings below 0.4 or with significant cross-loadings (i.e., loading differences between factors < 0.1) were systematically removed. After the first round of EFA, four items were excluded. In the second round, three additional items were excluded, followed by one item in the third round and two items in the fourth round. Therefore, a total of 10 items were excluded across four rounds of EFA, resulting in 32 items remaining.

After conducting EFA and excluding 10 items, the remaining 32 items were classified into two factors: 22 items in the first factor and 10 items in the second factor. To balance the number of items between the factors, we selected the top 10 items with the highest factor loadings from the first factor. For the second factor, the 10 items retained after EFA were ranked by their factor loadings. Consequently, 10 items were selected from each factor based on the highest loadings, resulting in a 20-item structure. Confirmatory factor analysis (CFA) was then conducted using these 20 items. Subsequently, to identify the model with the best fit, items with the highest factor loadings were selected from both factors, and models with different numbers of items per factor (e.g., eight items, seven items, and five items) were tested. The model with five items per factor demonstrated the highest model fit indices. Therefore, the final version of the Attitudes Toward Dreams Scale consisted of 10 items, with five items representing each factor.

Based on the findings from the EFA and CFA, factor scores were calculated as the mean of the respective five-item scores. No reverse-coded items were included. Model fit was evaluated using the goodness-of-fit index [27], the root mean square error of approximation [27], the chi-square minimum fit index, and Akaike's information criterion [28]. Acceptable model fit was defined as a goodness-of-fit index ≥ 0.90 and a root mean square error of approximation ≤ 0.08. Items with factor loadings of 0.4 or higher were retained, confirming that the scale appropriately measured the intended constructs. Internal consistency was assessed using Cronbach's alpha for each factor.

Statistical analyses were conducted using SPSS version 29.0.1 (IBM Corp., Armonk, NY), and confirmatory factor analysis was performed using SPSS Amos version 29 (IBM Corp., Armonk, NY). Additionally, a Bonferroni correction was applied for multiple comparisons, and the significance level for statistical analyses was set at p < 0.05.

 

## Results

Table 1 shows a distribution of participants recruited from eight regions of Japan. Japan comprises 47 prefectures, which are grouped into eight regions. The participant distribution closely reflects the actual population distribution, based on the 2020 national census. No significant skewing in sex or mean age was observed across the regions.

Table 2 shows the results of the descriptive statistics concerning the sleep- and mental health-related questionnaires used in this study.

Table 3 shows the distribution of dream-recall frequency for the total sample and based on sex. Since the number of choices in the surveys used in this study and Schredl's study [5] is different, they cannot be compared. Therefore, the results of both studies are shown in the same table for reference only.

Table 4 shows the newly developed Attitude Toward Dreams scale in this study and the standardized factor loadings for each scale item. We conducted an EFA (using the main factor method and oblique Promax rotation) on the 42-item scale to determine the number of factors from the scree plot and conceptual validity. Items with overlapping factors (factor loading difference < 0.1) were removed. A CFA was then conducted to confirm the model's fit, which resulted in two factors with five items each.

Fig 1 shows the path diagram of the results of the CFA of the 10-item Attitudes Toward-Dreams scale. The correlation coefficient between Factors 1 and 2 was 0.14 ($p < 0.01$).

Table 5 shows the correlations between the two factors of the Attitudes Toward-Dreams scale and various other scales, including dream-recall frequency, mid-arousal, RBDQ-J, STOP-Bang, restless legs syndrome, Athens Insomnia Scale, PSQI-J, SF-8, NEO-FFI, Japanese version of the Metamemory in Adulthood, and Dream Propensity Scale.

Factors 1 and 2 were significantly correlated with dream-recall frequency; the RBDQ-J; STOP-Bang; Athens Insomnia Scale; PSQI-J; the mental component summary of the SF-8; neuroticism and openness; the capacity, strategy, and locus of control subscales of the Japanese version of the Metamemory in Adulthood; and the bizarreness, impression, and activity subscales of the Dream Propensity Scale. However, all correlation coefficients were higher for Factor 1 than for Factor 2. Conversely, only the physical component summary of the SF-8 was significantly correlated with Factor 2.

Table 6 shows the mean scores and standard deviations for each element and Attitudes Toward-Dreams scale item, with a maximum score of 7. The mean score for Factor 1 was 3.59 ± 1.30, while that for Factor 2 was 4.29 ± 1.23. The mean score for Factor 1 was significantly lower than that for Factor 2 (t(1477) = 16.007, p < 0.001).

Fig 2 illustrates the age-related trends in Factor 1 by sex. A significant main effect of age was observed ($F(5, 1466) = 10.262$, $p < 0.001$), whereas the main effect of sex was marginally significant ($F(5, 1466) = 2.917$, $p = 0.088$). The

**Table 1. Number of Participants in Each Region of Japan Showing that Population Distributions are Similar to the 2020 National Census Data.**

| Region | Men | Women | Total | Mean Age ± SD | Male% | % | Actual% (2020) |
|---|---|---|---|---|---|---|---|
| Hokkaido | 39 | 49 | 88 | 51.49 ± 15.85 | 44.3 | 6.0 | 4.1 |
| Tohoku | 48 | 35 | 83 | 51.02 ± 17.65 | 57.8 | 5.6 | 6.8 |
| Kanto | 287 | 284 | 571 | 50.36 ± 16.59 | 50.3 | 38.6 | 34.6 |
| Chubu | 106 | 127 | 233 | 47.64 ± 16.04 | 45.5 | 15.8 | 16.8 |
| Kinki | 135 | 145 | 280 | 50.48 ± 17.02 | 48.2 | 18.9 | 17.7 |
| Chugoku | 37 | 32 | 69 | 49.61 ± 16.06 | 53.6 | 4.7 | 5.8 |
| Shikoku | 22 | 20 | 42 | 46.74 ± 16.46 | 52.4 | 2.8 | 2.9 |
| Kyushu | 54 | 58 | 112 | 48.26 ± 16.70 | 48.2 | 7.6 | 11.3 |
| Total | 728 | 750 | 1478 | 49.76 ± 16.59 | 49.3 | 100 | 100 |

Actual% (2020): The actual population distribution based on data from the 2020 national census. SD, standard deviation

**Table 2. Descriptive Statistics for Sleep- and Mental Health-related Questionnaires by Sex.**

| Scales | | Men | Women | Total |
|---|---|---|---|---|
| **RBDQ-J** | | 3.58 ± 2.68 | 3.68 ± 2.67 | 3.63 ± 2.68 |
| **STOP-Bang** | | 1.18 ± 1.15 | 0.88 ± 0.94 | 1.03 ± 1.06 |
| **RLS** | | 0.03 ± 0.17 | 0.03 ± 0.17 | 0.03 ± 0.17 |
| **AIS** | | 5.18 ± 4.51 | 5.19 ± 4.62 | 5.18 ± 4.56 |
| **PSQIG** | | 5.46 ± 3.16 | 5.71 ± 3.29 | 5.59 ± 3.23 |
| **Metamemory** | JMIA Change | 18.90 ± 5.71 | 18.15 ± 5.52 | 18.52 ± 5.62 |
| | JMIA Task | 36.74 ± 6.81 | 38.07 ± 6.53 | 37.42 ± 6.70 |
| | JMIA Capacity | 19.42 ± 4.59 | 20.06 ± 4.74 | 19.74 ± 4.67 |
| | JIMA Anxiety | 24.81 ± 5.79 | 26.40 ± 6.21 | 25.62 ± 6.06 |
| | JMIA Strategy | 17.25 ± 4.43 | 17.61 ± 4.51 | 17.43 ± 4.47 |
| | JMIA Locus of Control | 14.98 ± 2.77 | 15.47 ± 2.93 | 15.23 ± 2.87 |
| **Personality Characteristics** | NEO-FFI Neuroticism | 35.82 ± 7.25 | 37.65 ± 8.16 | 36.75 ± 7.77 |
| | NEO-FFI Agreeableness | 39.25 ± 5.11 | 41.17 ± 5.63 | 40.22 ± 5.46 |
| | NEO-FFI Conscientiousness | 36.78 ± 4.63 | 36.81 ± 4.94 | 36.79 ± 4.79 |
| | NEO-FFI Openness | 36.94 ± 4.51 | 37.81 ± 4.76 | 37.38 ± 4.66 |
| | NEO-FFI Extraversion | 33.59 ± 6.17 | 33.74 ± 6.36 | 33.67 ± 6.26 |
| **DP Scale** | DP Scale Bizarreness | 17.48 ± 4.22 | 17.39 ± 4.34 | 17.44 ± 4.28 |
| | DP Scale Evaluation | 15.69 ± 3.75 | 15.89 ± 3.59 | 15.79 ± 3.67 |
| | DP Scale Impression | 15.15 ± 4.30 | 15.62 ± 4.40 | 15.39 ± 4.35 |
| | DP Scale Activity | 11.65 ± 2.75 | 12.00 ± 2.41 | 11.83 ± 2.59 |
| **SF-8** | Physical Functioning | 49.72 ± 6.37 | 49.29 ± 6.96 | 49.50 ± 6.68 |
| | Role Physical | 49.99 ± 7.16 | 49.70 ± 7.50 | 49.84 ± 7.33 |
| | Bodily Pain | 49.46 ± 8.73 | 49.13 ± 8.74 | 49.29 ± 8.73 |
| | General Health | 50.49 ± 7.78 | 50.81 ± 7.64 | 50.65 ± 7.71 |
| | Vitality | 49.32 ± 7.41 | 49.68 ± 7.11 | 49.50 ± 7.26 |
| | Social Functioning | 49.88 ± 7.63 | 49.86 ± 7.72 | 49.87 ± 7.67 |
| | Role Emotional | 49.65 ± 7.07 | 49.32 ± 7.73 | 49.48 ± 7.41 |
| | Mental Health | 50.35 ± 7.21 | 48.96 ± 8.05 | 49.64 ± 7.68 |
| | Physical Component Summary | 48.53 ± 6.57 | 48.72 ± 7.32 | 48.62 ± 6.96 |
| | Mental Component Summary | 49.36 ± 7.24 | 48.65 ± 7.73 | 49.00 ± 7.50 |

Data are presented as mean and standard deviation.

RBDQ-J, Japanese version of the REM Sleep Behavior Disorder scale; RLS, restless legs syndrome; AIS, Athens Insomnia Scale; PSQIG, Pittsburgh Sleep Quality Index global score; DP, Dream Propensity; SF-8, Short Form-8; JMIA, Japanese version of the Metamemory in Adulthood Questionnaire; NEO-FFI, NEO Five-Factor Inventory

**Table 3. Dream-Recall Frequency.**

| Category | Number of samples | | | |
|---|---|---|---|---|
| | (N = 1,478) | (N = 728) | (N = 750) | (N = 2056) |
| | Total | Men | Women | Schredl (2020) |
| **Almost every morning** | 6.00% | 6.87% | 5.20% | 12.33% |
| **Several times a week** | 14.14% | 14.15% | 14.13% | 26.80% |
| **About once a week** | 14.48% | 13.19% | 15.73% | 18.68% |
| **About 2–3 times a month** | 14.01% | 12.23% | 15.73% | 15.13% |
| **About once a month** | 8.46% | 7.42% | 9.47% | 16.97% |
| **Less than once a month** | 16.37% | 17.17% | 15.60% | ----- |
| **Never** | 26.52% | 28.98% | 24.13% | 11.09% |

**Table 4. Standardized Factor Loadings of Items of the Attitudes Toward Dreams Scale.**

| Attitude toward dreaming | Rotated factor loadings | |
| --- | --- | --- |
| | Factor 1 | Factor 2 |
| I think that one can learn about oneself through dreams. | **0.867** | −0.055 |
| I think that one can improve one's life by understanding one's dreams. | **0.838** | −0.064 |
| I want to better understand the dreams I have had. | **0.838** | −0.086 |
| I think that dreams are very important. | **0.815** | −0.032 |
| I think that dreams reveal the dreamer's personality. | **0.783** | −0.037 |
| I do not take my dreams seriously. | −0.004 | **0.747** |
| Even if I have dreams, they cannot be helpful. | −0.007 | **0.708** |
| I think that dreams have no meaning. | −0.071 | **0.695** |
| Dreams are a meaningless product of the brain. | 0.035 | **0.690** |
| There is no need to analyze dreams. | −0.110 | **0.689** |

Based on the attitude toward dreams for each factor, Factor 1 was designated "meaning of dreams," while Factor 2 was designated "no meaning of dreams."

The goodness-of-fit index, root mean square error of approximation, AIC, and CMIN values were 0.962, 0.071, 330.73, and 288.73, respectively. Cronbach's alpha for Factor 1 was 0.905, and that for Factor 2 was 0.844. Several models were tested, and the model with 10 items (5 items for each factor) showed the smallest values. The 14-, 16-, and 20-item models yielded AICs of 817.57, 1149.85, and 1725.23, respectively. Similarly, CMIN values increased with more items, with the 10-item model having the lowest CMIN (288.73), followed by the 14-, 16-, and 20-item models (759.57, 1083.85, and 1870.02, respectively). This confirmed that as the number of items increased, the CMIN values also increased.

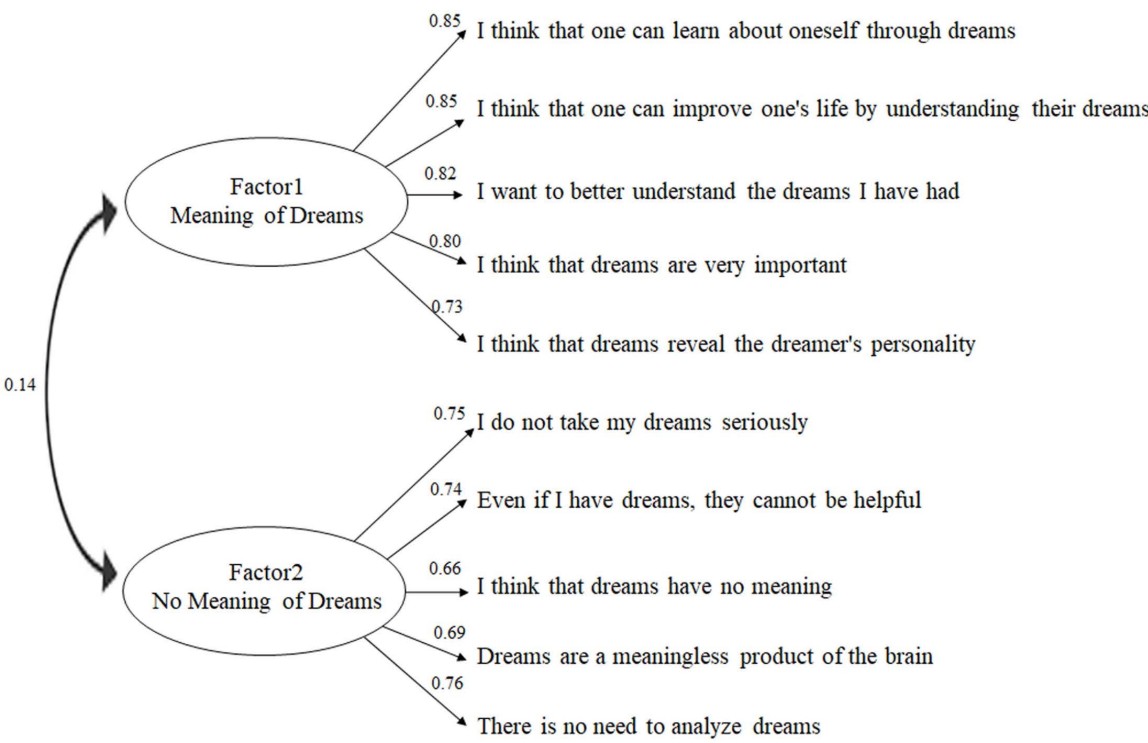

**Fig 1. Confirmatory Factor Analysis.** Confirmatory factor analysis of the 10-item Attitudes Toward Dreams Scale (N = 1,478), including standardized indices.

**Table 5. Correlations Between Factor Scores of the Attitudes Toward Dreams Scale and Other Scales.**

| Scales | Factor 1 | Factor 2 |
|---|---|---|
| **Dream-recall frequency** | 0.210** | 0.075** |
| **Awakenings during sleep** | 0.150** | 0.010 |
| **REM sleep behavior disorder (RBDQ-J)** | 0.273** | 0.150** |
| **Obstructive sleep apnea (STOP-Bang)** | 0.090** | 0.084** |
| **Restless legs syndrome** | 0.011 | 0.049 |
| **Insomnia (Athens Insomnia Scale)** | 0.148** | 0.137** |
| **PSQI-J (global Score)** | 0.132** | 0.072** |
| **Quality of life (SF-8)** | | |
| Physical Component Summary | −0.018 | −0.065* |
| Mental Component Summary | −0.099** | 0.098** |
| **NEO-FFI** | | |
| Neuroticism | 0.233** | 0.083** |
| Agreeableness | −0.031 | 0.049 |
| Conscientiousness | 0.165** | −0.034 |
| Openness | 0.170** | 0.164** |
| Extraversion | 0.013 | −0.003 |
| **Metamemory** | | |
| Change | −0.037 | 0.041 |
| Task | 0.133** | −0.038 |
| Capacity | 0.111** | 0.059* |
| Anxiety | 0.119** | 0.040 |
| Strategy | 0.131** | 0.103** |
| Locus of Control | 0.130** | 0.072** |
| **Dream Propensity Scale** | | |
| Bizarreness | −0.141** | −0.117** |
| Evaluation | 0.067** | 0.013 |
| Impression | 0.213** | 0.162** |
| Activity | 0.171** | 0.070** |

*$p<0.05$; **$p<0.01$

REM, rapid eye movement; RBDQ-J, Japanese version of the REM Sleep Behavior Disorder scale; PSQI-J, Japanese version of the Pittsburgh Sleep Quality Index; SF-8, Short Form-8; NEO-FFI, NEO Five-Factor Inventory

interaction between age and sex was not significant ($F(5, 1466) = 1.801$, $p = 0.110$). Despite the non-significant interaction, a simple main effects analysis revealed that the mean Factor 1 score significantly differed between men and women in their 20s. Additionally, among men, significant differences in mean scores were observed between participants in their 70s and those in all other age groups. Among women, significant mean score differences were observed between participants in their 60s and 70s and those in their 20s, 30s, 40s, and 50s.

Fig 3 illustrates the age-related trend in Factor 2 by sex. A significant main effect of age ($F(5, 1466) = 6.861$, $p < 0.001$) and sex ($F(1, 1466) = 5.949$, $p = 0.015$) was observed, although no significant interaction between the two variables was observed ($F(5, 1466) = 0.158$, $p = 0.978$).

Fig 4 shows the age trends for the two factors. Significant age-related differences were observed for both factors (Factor 1: $F(5, 1472) = 11.089$, $p < 0.001$; Factor 2: $F(5, 1472) = 7.188$, $p < 0.001$). Scores for both factors declined with age, with the steepest decline observed in the 70s age group. The mean Factor 1 score for the 70s age group significantly

**Table 6. Means and Standard Deviations of the 10-Item Attitudes Toward Dreams Scale (N = 1,478).**

| Items | Mean ± SD |
|---|---|
| **Factor 1** (Meaning of Dreams) | **3.59 ± 1.302** |
| I think that one can learn about oneself through dreams. | 3.47 ± 1.520 |
| I think that one can improve one's life by understanding one's dreams. | 3.44 ± 1.463 |
| I want to better understand the dreams I have had. | 3.75 ± 1.682 |
| I think that dreams are very important. | 3.67 ± 1.495 |
| I think that dreams reveal the dreamer's personality. | 3.61 ± 1.480 |
| **Factor 2** (No Meaning of Dreams) | **4.29 ± 1.227** |
| I do not take my dreams seriously. | 4.33 ± 1.604 |
| Even if I have dreams, they cannot be helpful. | 4.52 ± 1.468 |
| I think that dreams have no meaning. | 3.94 ± 1.604 |
| Dreams are a meaningless product of the brain. | 4.00 ± 1.630 |
| There is no need to analyze dreams. | 4.64 ± 1.504 |

The maximum score on the Attitudes Toward-Dreams scale is 7. SD, standard deviation.

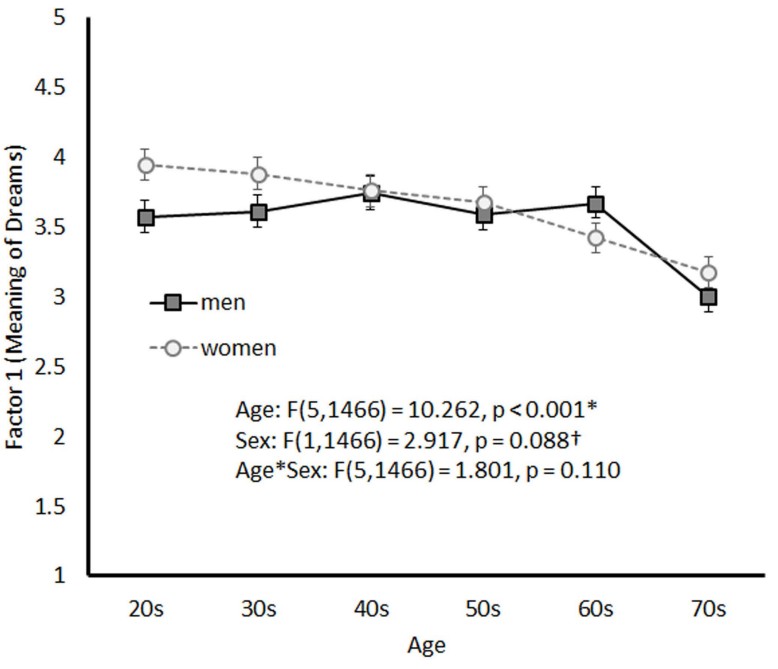

Age: $F(5,1466) = 10.262, p < 0.001$*
Sex: $F(1,1466) = 2.917, p = 0.088$†
Age*Sex: $F(5,1466) = 1.801, p = 0.110$

**Fig 2. Mean Scores for Factor 1 by Sex and Age Group.**

differed from the scores of all other age groups. The mean Factor 2 score for the 70s age group significantly differed from those of the other age groups, except for the 60s age group.

Fig 5 shows the age-related trend in dream-recall frequency by sex. No significant main effect of age or sex was observed (age: $F(5,1466) = 1.873, p = 0.171$; sex: $F(5,1466) = 0.616, p = 0.688$), although the interaction between the two variables was significant ($F(5,1466) = 4.739, p < 0.001$). Simple main effects analysis revealed significant sex differences in dream-recall frequency in the 20s and 30s age groups, while among women, significant differences was observed between the 30s age group and the 40s, 50s, and 60s age groups.

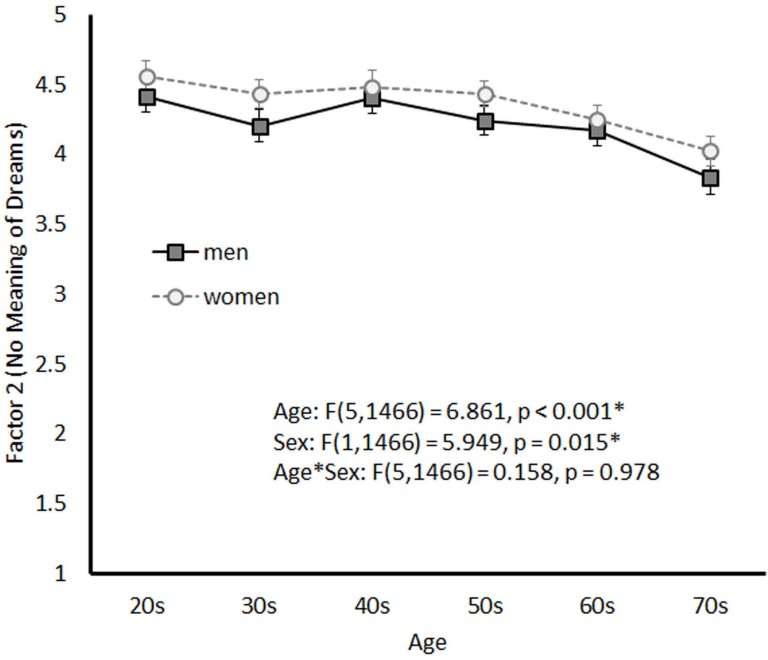

**Fig 3. Mean Scores for Factor 2 by Sex and Age Group.**

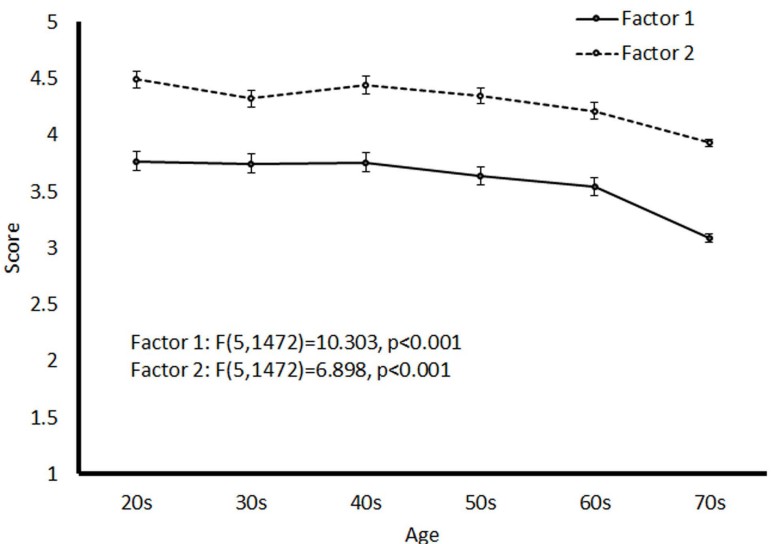

**Fig 4. Mean Score for Factors 1 and 2 in Each Age Group.**

## Discussion

This study aimed to develop a Japanese version of the Attitude Toward-Dreams scale and to examine the factors associated with Japanese participants' attitudes toward dreams. We incorporated several items from similar scales to those developed by Schredl et al. [3,4,6,9] and Kodama [11]. Kodama's scale was included because of its relevance to

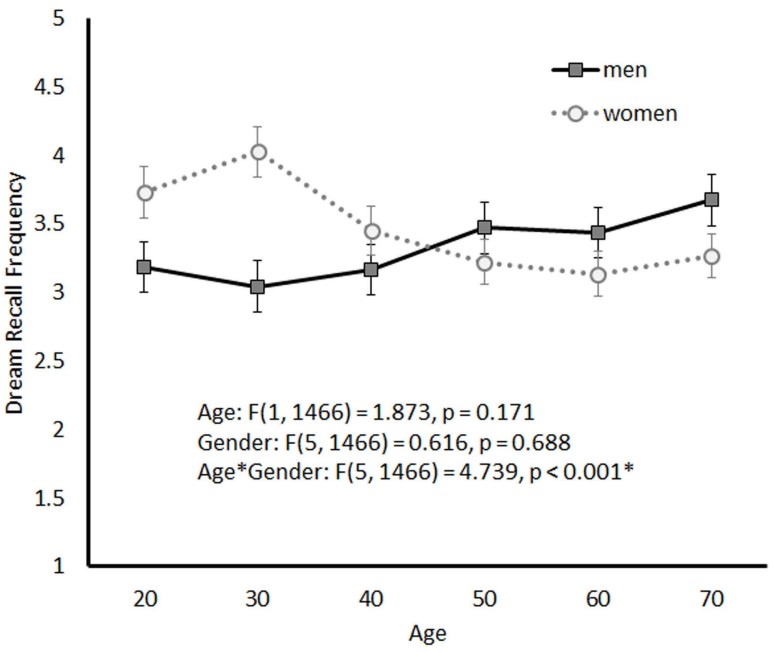

**Fig 5. Dream-Recall Frequency by Sex and Age Group.**

Japanese participants and its identification of more than two factors, suggesting potential cultural influences on attitudes toward dreams that warrant consideration.

Although the items were drawn from studies conducted in populations with different cultural backgrounds, the results revealed a simple two-factor structure similar to Schredl's findings. CFA confirmed the two-factor model as a reliable fit for the Japanese version of the Attitudes Toward Dreams Scale.

While Schredl referred to these factors as "positive" and "negative" attitudes, he noted that this naming could imply the two factors are merely opposites, which is not the case [6]. Factor 1 reflects an attitude of seeking "hidden" meanings in dreams, whereas Factor 2 represents a belief that dreams have no meaningful interpretation. Accordingly, we named Factor 1 "meaning of dreams" and Factor 2 "no meaning of dreams."

We investigated the relationship between the two factors and several sleep disorder variables, along with measures of quality of life, personality, memory, and dream characteristics. Significant correlations were identified, with Factor 1 showing the strongest associations. Factor 1 showed significant correlations with many items; among them, correlation coefficients above 0.2 were observed for dream-recall frequency ($r = 0.210$), RBD score ($r = 0.273$), neuroticism ($r = 0.233$), and dream impression ($r = 0.213$). However, Factor 2 did not show correlation coefficients exceeding 0.2.

Among the four items that showed correlation coefficients above 0.2 with Factor 1, all but neuroticism were primarily related to dream or REM sleep variables. Previous studies [2,29] have also revealed a significant positive correlation between neuroticism and positive attitudes toward dreams. Attitudes toward dreams have been studied with respect to factors such as dream-recall frequency, sex, age, ethnicity, and education level [4,30–32]; however, the relationship with sleep disorders has not been investigated, except for that with nightmares.

Our results suggest that attempting to find meaning in dreams may be associated with unhealthy sleep patterns. However, it is also possible that poor sleep and negative dream experiences increase interest in dream interpretation. Future intervention studies are needed to clarify these causal relationships.

The mean scores for Factors 1 and 2 showed a significant decline with age, suggesting that interest in dreams decreases with age. This trend was particularly evident among individuals in their 60s and 70s. Similar declines in interest in interpreting the meanings of dreams have been reported previously [4,10]. Although not shown in the results, we also analyzed age-group differences in the relationships between the factors and various items. Notably, the relationship between Factor 1 and RBD was relatively stronger in older adults (60s and 70s) than in that in younger participants (20s: r = 0.207, 30s: r = 0.207, 40s: r = 0.225, 50s: r = 0.195, 60s: r = 0.265, 70s: r = 0.307). This suggests that maintaining a high interest in dreams, even in old age, may indicate susceptibility to RBD. Overall, focusing on attitudes toward dreams may aid in the early detection of sleep disorders.

Schredl [2] also found that interest in dreams was a significant factor in explaining sex differences in dream-recall frequency. In this study, sex differences between Factor 1 and dream-recall frequency were limited to younger age groups (20s and 30s); therefore, differences in dream-recall frequency might become evident when investigated in younger participants.

## Limitations of the study

This study had some limitations. First, because data were collected through an online survey, many participants were excluded due to inconsistencies in their PSQI responses, suggesting possible misunderstandings of the questions or inattentive answering. In this study, since the correlation noted between Factor 1 and RBD was weak, its implications may be limited. Furthermore, previous studies have not identified a link between the tendency to find meaning in dreams and RBD, and the underlying mechanism for this relationship remains unclear. Therefore, we believe further detailed investigation is needed. While we investigated the relationship between attitudes toward dreams and mental health, we did not establish a causal relationship. Therefore, it remains unclear whether attitudes toward dreams influence mental health, whether mental health influences attitudes toward dreams, or whether other factors affect both. Moreover, participants in this study were not specifically screened for psychiatric conditions; therefore, the potential influence of such conditions on the results cannot be ruled out. Future studies should consider psychiatric conditions during participant recruitment to better understand their possible impact on dream recall. Additionally, both the EFA and CFA were conducted using the same dataset. While this approach was deemed appropriate for the initial development of the scale, further validation is necessary. Future studies should aim to replicate the identified factor structure using independent samples to establish the generalizability and stability of the scale across diverse populations.

## Conclusion

In this study, a new Attitude Toward Dreams scale was developed, consisting of two factors: "meaning of dreams" and "no meaning of dreams," each comprising five items. The results confirmed that this two-factor structure is consistent among Japanese participants, aligning with findings from studies conducted in other countries. The results also suggest that attitudes toward finding meaning in dreams may be associated with poorer sleep quality and lower quality of life, although the causal relationship remains unclear. In addition, we observed a gradual decline in interest in dreams with increasing age. The Attitude Toward Dreams scale developed in this study is less burdensome for respondents due to its fewer items. Additionally, the scale's potential to predict sleep disorders may be useful for screening sleep health.

## Supporting information

**S1 Data. Raw dataset including individual item responses and factor scores on the Japanese version of the Attitudes Toward Dreams Scale, used for statistical analyses reported in the study.**
(CSV)

**S2 File. Full version of the Japanese Attitudes Toward Dreams Scale, including original Japanese items, their English translations, response format, and usage permissions. This file also includes information about copyright and modification conditions.**
(DOCX)

## Acknowledgments

We thank everyone who contributed to the completion of this study.

## Author contributions

**Conceptualization:** Kenta Nozoe, Kazuhiko Fukuda, Takamasa Kogure, Shoichi Asaoka.

**Data curation:** Shinya Okuyama, Kenta Nozoe, Kazuhiko Fukuda.

**Formal analysis:** Kenta Nozoe, Kazuhiko Fukuda.

**Funding acquisition:** Takamasa Kogure.

**Investigation:** Shinya Okuyama, Kenta Nozoe, Kazuhiko Fukuda, Takamasa Kogure, Shoichi Asaoka.

**Methodology:** Kenta Nozoe, Kazuhiko Fukuda, Takamasa Kogure, Shoichi Asaoka.

**Supervision:** Kazuhiko Fukuda, Takamasa Kogure, Shoichi Asaoka.

**Visualization:** Shinya Okuyama, Kazuhiko Fukuda.

**Writing – original draft:** Shinya Okuyama, Kenta Nozoe, Kazuhiko Fukuda.

**Writing – review & editing:** Shinya Okuyama, Kenta Nozoe, Kazuhiko Fukuda, Takamasa Kogure, Shoichi Asaoka.

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
