## [Decision Letter · Decision Letter 0]

Dear Dr. Okuyama,

Thank you for submitting your manuscript to PLOS ONE. After careful consideration, we feel that it has merit but does not fully meet PLOS ONE’s publication criteria as it currently stands. Therefore, we invite you to submit a revised version of the manuscript that addresses the points raised during the review process.

After careful consideration of the reviewers' reports, it has been determined that the manuscript requires major revisions before it can be considered for publication.

We kindly ask you to respond to all the issues raised by the two referees, paying particular attention to addressing the methodological concerns they have highlighted. Furthermore, we recommend that you thoroughly revise the discussion sections to ensure clarity, consistency, and alignment with the data presented.

We look forward to receiving your revised manuscript.

Kind regards,

Serena Scarpelli

Academic Editor

PLOS ONE

Journal Requirements:

“This research was funded by PARAMOUNT BED CO., LTD.”

“Dr. Takamasa Kogure belongs to PARAMOUNT BED CO., LTD., however, their interests are not competing with the purpose of this study.  “

4.  Please include captions for your Supporting Information files at the end of your manuscript, and update any in-text citations to match accordingly. Please see our Supporting Information guidelines for more information: http://journals.plos.org/plosone/s/supporting-information .

Reviewers' comments:

Reviewer's Responses to Questions

**Comments to the Author**

1. Is the manuscript technically sound, and do the data support the conclusions?

Reviewer #1: Partly

Reviewer #2: Yes

2. Has the statistical analysis been performed appropriately and rigorously?

Reviewer #1: No

Reviewer #2: Yes

3. Have the authors made all data underlying the findings in their manuscript fully available?

Reviewer #1: Yes

Reviewer #2: Yes

4. Is the manuscript presented in an intelligible fashion and written in standard English?

Reviewer #1: Yes

Reviewer #2: Yes

Reviewer #1: Overall, the paper is well written and the authors' effort to incorporate and validate Western instruments into Japanese culture is interesting and deserves to be supported because it generally improves scientific progress in general. However, the study has some important methodological problems that need to be corrected or adequately explained. In an attempt to incorporate instruments into Japanese culture, the authors have made an effort that may have been too optimistic and with methods that statistically favor obtaining associations, such as performing a CFA followed by an EFA. The fit indices employed by the authors are not particularly stringent. This leads them to exaggerated conclusions, including the title.

Reviewer #2: Review PONE-D-24-51943

Overall comment:

Thank you for the opportunity to read and review this paper. It is an interesting paper, where a scale about people’s attitudes toward dreams has been developed and tested in a Japanese population. It is of interest and relevance for sleep medicine but could do with some clarifications to make it easier for the reader to follow.

Abstract:

The aim of the study is described differently in the abstract compared to in the paper. Also, in the aim described in the abstract, the authors write about relevant factors influencing attitudes toward dreams. What does relevant factors mean in this context? It would be clearer if the authors specified the factors.

Introduction:

Page 3, line 54-56: here the aim is presented once again, adding a possible development of a multidimensional scale (which the one in the abstract does not say). Since scale development seems to have been the general idea from the start, this should be added to the aim in the abstract and I also think that ‘if necessary’ (page 3, line 55) can be omitted.

Page 3, line 57-62:

Consider moving this text to the Materials and methods section instead. It seems more to describe what the authors have done rather than provide a rationale for the study.

Materials and methods

Participants and data collection

Recruitment and data collection happened in only 5 days, did participants provide their written, informed consent in this time as well, or was consent collected before sending out the survey?

Did the authors consider any reminders?

How were the participants selected, was the sample stratified in some way?

Attitudes toward dreams

Page 5

I’d like a more thorough description of the scale development. How did the author decide on what items to pick from each of the previous questionnaires?

What was the motive for changing the response-alternatives from the original scale?

Analysis

Page 5

As far as I can tell, the analysis seems to be alright, but perhaps the authors would consider starting with describing what analyses they will use and stepwise describe the scale development. This could be a way to more clearly describe how they chose items from the other questionnaires to form a new one.

Results

It would have been nice with a table presenting the participants of the study, describing the distribution between sex, mean age, geographical area, mean score of the scales used.

Page 6, Table 1

This table shows a comparison between the current study sample and a previous study by Schredl, however, it seems that there were different number of response-alternatives in the studies, or why is it a blank in the column describing the data of Schredl? How should that be interpreted, that is, with fewer options, how would they have responded compared to the present study? I don’t think you can compare straight across the samples given these circumstances.

Page 7, Table 2

Just to be clear, table 2 shows the scale that was developed in the current study. Maybe this could be clearly stated in the text.

Page 8, line 153

Just a minor comment, the authors should not present Table 4 in the text before presenting Table 3 in the text.

Page 9, line 167-169

There are several measures mentioned in the text below the table, but where can I see the data from those measures (e.g. blood pressure, BMI, age, neck circumference, gender). Maybe this could be added to a table describing the participants of the study?

Several of the correlations presented in table 3 were statistically significant, however the correlations were weak in general, which is important to remember when considering the findings.

Page 10, Table 4

Could the authors please add the maximum score for each item, just so that the reader can evaluate what the mean-values indicate.

Page 11, line 212

The text says there is a difference between sex for dream frequency, but shouldn’t it be dream-recall frequency? I believe it is only subjective data collected, i.e. what the persons remember regarding their dreams, no EEG that can tell objectively if and how much they have dreamt.

The term dream frequency occurs again in page13, line 260, 261 and 262.

Discussion

Page 12, line 219

Here is a third version of the aim, stating that the purpose of the study was to develop a scale. Maybe the authors could agree on one of the three versions of the aim and then use that throughout the paper for clarity.

Page 12, line 235

Here the authors discuss the correlations between the Attitudes toward dreams scale and other scales, pointing out that there are statistically significant correlations, but it should be discussed that they are weak and what that implies in terms of interpreting the clinical significance of the correlations.

Page 13, line 256-257

It is suggested that having a high interest in dreams in older age may indicate susceptibility to RBD. How so? In RBD vivid, frightful dreams may occur, but not always remembered by the dreamer. Another feature is that the normal REM-sleep muscle atony is disconnected, and the person may ‘live’ their dream. I have never heard that remembering one’s dreams or trying to find meaning in them would be related to RBD. The authors may want to add references that supports their statement or change it.

Limitations of the study

Page 13, line 267

I do not understand why the survey being online would have caused participants to misunderstand questions, if the authors please would elaborate on that. I think that if a question is misunderstood it is about the wording and that would be the same for both paper questionnaire and online surveys.

Page 14, line 272

The results may have been influenced by cultural factors of Japan. I do not see this as a limitation, as the authors clearly state that they wish to investigate attitudes toward dreams in Japanese men and women. (I understood it as to really consider possible cultural factors, that may be specific for this population). However, it would be interesting if the authors presented some cultural differences found through the study and compare these to previous studies in other populations.

**Do you want your identity to be public for this peer review?** For information about this choice, including consent withdrawal, please see our Privacy Policy

Reviewer #1: No

Reviewer #2: No

---

## [Author Response · Author response to Decision Letter 1]

12 Mar 2025

Reviewer #1: Overall, the paper is well written and the authors' effort to incorporate and validate Western instruments into Japanese culture is interesting and deserves to be supported because it generally improves scientific progress in general. However, the study has some important methodological problems that need to be corrected or adequately explained. In an attempt to incorporate instruments into Japanese culture, the authors have made an effort that may have been too optimistic and with methods that statistically favor obtaining associations, such as performing a CFA followed by an EFA. The fit indices employed by the authors are not particularly stringent. This leads them to exaggerated conclusions, including the title.

【Response】We thank you for your insightful review of our manuscript and valuable comments. It seems there was a misunderstanding about the order of exploratory factor analysis (EFA) and confirmatory factor analysis (CFA). In this study, we first performed EFA to explore the factor structure based on the data. After that, we conducted CFA to verify whether the explored factor structure fits the data. This approach follows the standard procedure in factor analysis, and by confirming the factor structure explored through EFA with CFA, we can obtain more robust results. Therefore, we believe that we have not used a statistically biased method regarding the order of EFA and CFA. We have added the analysis procedure to the “Methods” section.

―――――――――――――――――――――――――――――――――――――

Reviewer #2: Review PONE-D-24-51943

Overall comment:

Thank you for the opportunity to read and review this paper. It is an interesting paper, where a scale about people’s attitudes toward dreams has been developed and tested in a Japanese population. It is of interest and relevance for sleep medicine but could do with some clarifications to make it easier for the reader to follow.

Abstract:

The aim of the study is described differently in the abstract compared to in the paper. Also, in the aim described in the abstract, the authors write about relevant factors influencing attitudes toward dreams. What does relevant factors mean in this context? It would be clearer if the authors specified the factors.

【Response】We thank you for your insightful review of our manuscript and valuable comments. Our point-by-point responses are provided below.

We appreciate your suggestion to clarify the aim of our study and specify the "relevant factors" mentioned in the abstract.

In response to your comment, we have revised the abstract to explicitly specify the factors examined in our study. These include: dream attitudes (Factor 1: meaning of dreams, Factor 2: no meaning of dreams), dream-recall frequency, various sleep-related variables (e.g., Pittsburgh Sleep Quality Index, sleep apnea, restless legs syndrome, insomnia, and rapid eye movement sleep behavior disorder), personality traits, and quality of life. We believe that this revision provides a clearer understanding of the factors studied and aligns the aim of the study more closely with the content of the paper. We hope this revision addresses your concern.

Introduction:

Page 3, Line 54-56: here the aim is presented once again, adding a possible development of a multidimensional scale (which the one in the abstract does not say). Since scale development seems to have been the general idea from the start, this should be added to the aim in the abstract and I also think that ‘if necessary’ can be omitted.

【Response】To ensure consistency between the Abstract and the text regarding the purpose of the study, we should have explicitly stated the multidimensional development of the “Attitude Toward Dreams Scale” in the Abstract as well. Therefore, we have added the sentence “We aimed to develop a dream attitude scale” to the Abstract, and we have deleted the phrase “as needed”. The modification is as follows:

Page 2, Line 17-22: “Therefore, in this study, we aimed to identify the relevant factors influencing attitudes toward dreams among Japanese participants. These factors included dream attitudes, dream-recall frequency, various sleep-related variables (Pittsburgh Sleep Quality Index, sleep apnea, restless legs syndrome, insomnia, and rapid eye movement sleep behavior disorder), personality traits, and quality of life. In addition, we also aimed to develop a new dream attitude scale."

Page 3, Line 56-58: “Therefore, this study aimed to identify the factors specific to Japanese participants and, if necessary, develop a multidimensional scale for measuring “attitudes toward dreams” by creating new questionnaire items based on existing dream attitude scales.”

Page 3, Line 57-62: Consider moving this text to the Materials and methods section instead. It seems more to describe what the authors have done rather than provide a rationale for the study.

【Response】Indeed, the information on dream recurrence frequency, age, sleep habits, personality traits, and metamemory should be explained as part of the research methodology, and it would be more appropriate to move it to the Materials and Methods section as a description of the methodology rather than a description of the purpose of the study. Therefore, this section has been moved to the Materials and Methods section and the relevant section has been revised (Page 3, Lines 61-67).

Materials and methods

Participants and data collection

Recruitment and data collection happened in only 5 days, did participants provide their written, informed consent in this time as well, or was consent collected before sending out the survey?

【Response】Data collection for this study was contracted to a web-based research firm, Cross Marketing, Inc. Because the survey was conducted online, consent was confirmed online rather than in writing. At that time, those who did not consent were not allowed to respond to the survey.

Did the authors consider any reminders?

【Response】The data collection for this study was outsourced to Cross Marketing Inc, a web survey company. We are unaware of whether reminders were sent. However, considering that the data was collected within a 5-day period, it seems unlikely that reminders were issued.

How were the participants selected, was the sample stratified in some way?

【Response】The participants in this study were randomly selected from individuals who had previously participated in another survey conducted by Cross Marketing (a web survey company) within the past year. The sample included men and women in their 20s to 70s, and the survey URL was closed once the target sample size was reached for participants who completed the entire questionnaire. However, to account for potential data loss during the data cleaning process, we collected approximately 10% more data than the intended sample size. In the end, Cross Marketing delivered data from 1,680 valid responses, randomly selected from the pool. The breakdown of the 1,680 participants was specified by us, with 140 men and 140 women in each age group from the 20s to 70s.

Attitudes toward dreams

Page 5

I’d like a more thorough description of the scale development. How did the author decide on what items to pick from each of the previous questionnaires?

【Response】In previous studies, the standard factor structure of the Attitudes Toward Dreaming Scale was considered to be based on “positive and negative attitudes toward dreams,” so the Mannheim Dream Questionnaire (2014), Attitudes toward Dreaming (2019), and Attitudes Toward Dreams Scale (2010) were employed. Furthermore, we wondered if we could create an Edogawa University version of the original Attitudes Toward Dreaming scale by adding and validating Dr. Kodama's Dream Attitudes Scale (2000), which had published a factor structure other than “positive and negative attitudes toward dreams” in this study.

What was the motive for changing the response-alternatives from the original scale?

【Response】In the attitude questionnaire toward dreams conducted in this study, a 7-point scale was used. Schredl used a 5-point scale and Kodama used a 4-point scale, so it was necessary to unify them in this study. Weng (2004) reported that "whether language labels are provided for all categories or only for the endpoints, having fewer response categories results in lower reliability." Additionally, Krosnick & Presser (2010) provided comprehensive guidelines for survey questionnaire and scale design, stating that a 7-point scale "allows respondents' opinions to be more accurately reflected by including a middle option." However, neither of these studies concluded that the 7-point scale is superior to other methods. Nevertheless, in this study, in order to enhance the reliability of the survey results, the number of response categories was increased.

Additionally, the following two sentences were added under “Attitudes toward dreams” in the Materials and Methods section.

Page 5, Lines 106-108: “Since these studies used the five-point scale [2, 4, 9] or the four-point scale [11], the questionnaire in this study was standardized to a seven-point scale.”

　Page 5, lines 109-115: “Weng [25] reported that "whether language labels are provided for all categories or only for the endpoints, having fewer response categories results in lower reliability." Additionally, Krosnick and Presser [26] provided comprehensive guidelines for survey and scale design, noting that a seven-point scale "can more accurately reflect respondents' opinions by including a middle option." However, neither of these studies concluded that the seven-point scale is superior to other methods. Nevertheless, in this study, a seven-point scale was used in the survey on attitudes toward dreams to enhance the reliability of the results.”　

Analysis

Page 5

As far as I can tell, the analysis seems to be alright, but perhaps the authors would consider starting with describing what analyses they will use and stepwise describe the scale development. This could be a way to more clearly describe how they chose items from the other questionnaires to form a new one.

【Response】In this study, we developed a new “Attitude Toward Dreams scale" based on four previous studies. The flow of analysis on the results of the questionnaire regarding attitudes toward dreams has been added to the main text.

Pages 5-6, Lines 118-129: “The results of the questionnaire on the Attitude Toward Dreams scale were first subjected to exploratory factor analysis (EFA) to extract the factor structure. Parallel analysis and eigenvalue criteria were used to determine the number of factors, retaining items with factor loadings of 0.4 or higher. As a result, several factors were clearly extracted, and the scale structure was conceptually clarified. Based on the EFA findings, confirmatory factor analysis (CFA) was conducted to assess the goodness of fit of the scale. In the CFA, model fit indices (the goodness of fit was assessed using the goodness-of-fit index, root mean square error of approximation, chi-square minimum fit index (CMIN), and Akaike’s information criterion (AIC)) were used to verify how well the structure fit the data. Items with factor loadings of 0.5 or higher were selected, confirming that the scale appropriately measured the intended constructs. For the evaluation of reliability and validity, Cronbach's alpha was used for each factor. Additionally, Bonferroni correction was applied for multiple comparisons to confirm statistical significance.”

Results

It would have been nice with a table presenting the participants of the study, describing the distribution between sex, mean age, geographical area, mean score of the scales used.

【Response】I have created Table 1, titled “Number of participants in each area of Japan.”

The accompanying description for Table 1 has also been revised:

Pages 6-7, Lines 145-148: “Table 1 shows a distribution of participants recruited into the study from eight areas of Japan. Japan has 47 prefectures, which are summarized into eight areas. The population distribution in this study is similar to the actual distribution. There was no prominent skewing in terms of sex or mean ages among these areas.”

Page 6, Table 1

This table shows a comparison between the current study sample and a previous study by Schredl, however, it seems that there were different number of response-alternatives in the studies, or why is it a blank in the column describing the data of Schredl? How should that be interpreted, that is, with fewer options, how would they have responded compared to the present study? I don’t think you can compare straight across the samples given these circumstances.

【Response】As stated in the Methods section, this study used a 7-point scale survey. Therefore, as you pointed out, it was not appropriate to compare the results of this study with those of Schredl. However, we wanted to show the difference between the two sides, as their results were very different. I have added a note in the text clarifying that the inclusion of Schredl’s results in the table was solely for reference.

I have added the following explanation regarding Table 2.

Page 7, Lines 153-156: “Table 2 shows the distribution of dream-recall frequency for the total sample and based on sex. Since the number of choices in the surveys used in this study and Schredl’s study [5] is different, they cannot be compared. Therefore, the results of both studies are shown in the same table for reference only.”

Page 7, Table 2

Just to be clear, table 2 shows the scale that was developed in the current study. Maybe this could be clearly stated in the text.

【Response】The text describing Table 3 “presents the newly developed Attitude to Dreams scale developed in this study and the standardized factor loadings of the items on the scale.” The text has been modified as follows.

Page 8, Lines 150-161: “Table 3 shows the newly developed Attitude Toward Dreams scale in this study and the standardized factor loadings for each scale item.”

Page 8, line 153

Just a minor comment, the authors should not present Table 4 in the text before presenting Table 3 in the text.

【Response】Table 1 is newly added, and thus, the previous Table 3 has been changed to Table 4 and Table 4 to Table 5. I fixed the order in which Tables are presented.　

In addition, an explanation of the results shown in Table 5 is noted below.　

Page 11, Lines 202-205: “Table 5 shows the mean scores and standard deviations for each element and Attitude Toward Dreams scale item, with a maximum value of 7. The mean score for Factor 1 was 3.59 ± 1.30, while that for Factor 2 was 4.29 ± 1.23. The mean score for Factor 1 was significantly lower than that for Factor 2 (t (1477) = 16.007, p < 0.001).”

Page 9, line 167-169

There are several measures mentioned in the text below the table, but where can I see the data from those measures (e.g. blood pressure, BMI, age, neck circumference, gender). Maybe this could be added to a table describing the participants of the study?

【Response】We did not investigate “high blood pressure,” “body mass index,” or “neck circumference” as you pointed out. It was our mistake to describe this. I am very sorry about that. Information on age and gender differences is provided in Table 1.Several of the correlations presented in table 3 were statistically significant, however the correlations were weak in general, which is important to remember when considering the findings.

【Response】We are aware of the overall low correlation in the data of this study as per your comment. We indicate this in the text.

Page 10, Table 4

Could the authors please add the maximum score for each item, just so that the reader can evaluate what the mean-values indicate.

【Response】I have indicated a maximum score of “7” in the legend of the new Table 5 (Page 12, Line 209).

Page 11, line 212

The text says there is a difference between sex for dream frequency, but shouldn’t it be dream-recall frequency? I believe it is only subjective data collected, i.e. what the persons remember regarding their dreams, no EEG that can tell objectively if and how much they have dreamt.

The term dream frequency occurs again in page13, line 260, 261 and 262.

【Response】As you mentioned, I believe “dream-recall frequency” is the more appropriate exp

---

## [Decision Letter · Decision Letter 1]

Dear Dr. Okuyama,

Thank you for submitting your manuscript to PLOS ONE. After careful consideration, we feel that it has merit but does not fully meet PLOS ONE’s publication criteria as it currently stands. Therefore, we invite you to submit a revised version of the manuscript that addresses the points raised during the review process.

Some minor issues should be addressed before acceptance.

We look forward to receiving your revised manuscript.

Kind regards,

Serena Scarpelli

Academic Editor

PLOS ONE

Journal Requirements:

Reviewers' comments:

Reviewer's Responses to Questions

**Comments to the Author**

Reviewer #1: All comments have been addressed

Reviewer #3: (No Response)

2. Is the manuscript technically sound, and do the data support the conclusions?

Reviewer #1: Yes

Reviewer #3: Yes

3. Has the statistical analysis been performed appropriately and rigorously?

Reviewer #1: Yes

Reviewer #3: Yes

4. Have the authors made all data underlying the findings in their manuscript fully available?

Reviewer #1: (No Response)

Reviewer #3: Yes

5. Is the manuscript presented in an intelligible fashion and written in standard English?

Reviewer #1: Yes

Reviewer #3: Yes

Reviewer #1: I noticed that you have included the analysis procedure at the end of the 'Methods' section. While this is adequate, for future submissions, I would recommend adding a dedicated section that provides a detailed explanation of the statistical technique employed. Additionally, it may be beneficial to suggest the use of distinct data samples for the Exploratory Factor Analysis (EFA) and Confirmatory Factor Analysis (CFA)

Reviewer #3: The article is interesting and seems to have improved since the first version. However, there is still need for improvement. Morevore, I noted that the authors did not aswer to the reviewers point by point:

1) It is necessary to specify the aim of the study, because the authors still report three different versions in the text.

Page 2, lines 17-18 " we aimed to identify theidentify the relevant factors influencing attitudes toward dreams among Japanese participants";

Page 3, lines 56-58, "this study aimed to identify the factors specific to Japanese participants and, if necessary, develop a multidimensional scale for measuring “attitudes toward dreams” by creating new questionnaire items based on existing dream attitude scales”;

Page 13, Line 246, "the purpose of this study was to develop a Japanese version of the attitudes-toward-dreams scale."

2) It is not clear how you selected the final sample. How did you get from 1680 to 1478 participants? What do you mean in line 74 “who provided consistent responses about sleep-related factors?" Please explain the inclusion and exclusion criteria in detail.

2.1) Also, if participants were contacted in a previous data collection, you need to make this explicit in the main text. It should also be made clear whether all of the data collected was collected during the reporting period (from 18 March 2022 to 22 March 2022) or whether some of the information was obtained from the previous data collection.

2.2) Has the presence of other medical and/or psychiatric conditions been checked at recruitment? If not, this should be made explicit in the boundaries, as the presence of some of these conditions may affect dream recall.

3) The description of how you scaled the Attitudes toward dreams scale should be included in the main text.

Result

4) Age-related ds should also be added to Table 1. Also, Actual% 2020 refers to what? You should describe this in the heading.

5) A table with descriptive analyses of any other measures you collected in the survey should be included.

6) Table 4 still shows several measures such as blood pressure, BMI, age, neck circumference, gender, but not the data from these measures?

**Do you want your identity to be public for this peer review?** For information about this choice, including consent withdrawal, please see our Privacy Policy

Reviewer #1: No

Reviewer #3: No

---

## [Author Response · Author response to Decision Letter 2]

11 May 2025

PONE-D-24-51943R1

Creating a Japanese version of the Attitudes Toward Dreams scale: Attitude toward dreams may predict sleep disorders

[Responses to Reviewers' comments/questions]

Reviewer #1: I noticed that you have included the analysis procedure at the end of the 'Methods' section. While this is adequate, for future submissions, I would recommend adding a dedicated section that provides a detailed explanation of the statistical technique employed. Additionally, it may be beneficial to suggest the use of distinct data samples for the Exploratory Factor Analysis (EFA) and Confirmatory Factor Analysis (CFA)

Response: Thank you for your suggestion to include a separate section with a more detailed explanation of the statistical methods used. To address your comments, we added a statement to the limitations section clarifying that both the exploratory factor analysis (EFA) and confirmatory factor analysis (CFA) were conducted using the same dataset. We will adopt your recommended approach for data analysis and reporting in future, more advanced versions of this work. The following text has been added to the limitations section (page 19, lines 361–365).

“Additionally, both the EFA and CFA were conducted using the same dataset. While this approach was deemed appropriate for the initial development of the scale, further validation is necessary. Future studies should aim to replicate the identified factor structure using independent samples to establish the generalizability and stability of the scale across diverse populations.”

Reviewer #3: The article is interesting and seems to have improved since the first version. However, there is still a need for improvement. Moreover, I noted that the authors did not answer the reviewers' point-by-point:

1) It is necessary to specify the aim of the study, because the authors still report three different versions in the text.

Page 2, lines 17-18 " we aimed to identify the relevant factors influencing attitudes toward dreams among Japanese participants";

Page 3, lines 56-58, "this study aimed to identify the factors specific to Japanese participants and, if necessary, develop a multidimensional scale for measuring “attitudes toward dreams” by creating new questionnaire items based on existing dream attitude scales”;

Page 13, line 246, "the purpose of this study was to develop a Japanese version of the attitudes-toward-dreams scale."

Response: Thank you for pointing out the inconsistencies in the description of the study’s aim. To address this issue and improve clarity, we revised the text to provide a unified and consistent statement of the aim of the study throughout the manuscript. The statements have been revised as follows.

・Page 2, lines 17–19: “This study aimed to develop a Japanese version of a scale to assess attitudes toward dreams and to examine factors related to Japanese participants' attitudes toward dreams.”

・Page 3, line 55–57: “Therefore, this study aimed to develop a Japanese version of a scale to assess attitudes toward dreams and to examine the factors associated with Japanese participants' attitudes toward dreams.”

・Page 16, lines 303–304: “This study aimed to develop a Japanese version of the Attitude Toward-Dreams scale and to examine the factors associated with Japanese participants’ attitudes toward dreams.”

2) It is not clear how you selected the final sample. How did you get from 1680 to 1478 participants? What do you mean in line 74 “who provided consistent responses about sleep-related factors?" Please explain the inclusion and exclusion criteria in detail.

Response: Thank you for pointing this out. To clarify the sample selection process and address your recommendation to describe the inclusion and exclusion criteria, we added the following text to the Methods section (page 4, lines 74–87):

“To assess sleep-related parameters, we used the Japanese version of the Pittsburgh Sleep Quality Index (PSQI-J). While Cross Marketing Inc. ensured that all questionnaire items were completed (i.e., there were no missing responses), the data from 202 participants were excluded from analysis because of internally inconsistent responses regarding bedtime, wake-up time, and sleep duration, which made it impossible to accurately calculate their global PSQI scores. Responses were considered inconsistent if they included, for example: (1) a reported sleep latency of 120 min, a sleep duration of 8 h, and a total time in bed of only 7 h (i.e., logically impossible); (2) a bedtime of 10:00 PM and a wake-up time of 8:00 AM, but a sleep duration of only 2 h without any reported nighttime awakenings; or (3) a sleep duration that exceeded the total time in bed (e.g., 8.5 h of sleep during 6 h in bed). These types of discrepancies, likely due to input errors or misunderstandings of the questionnaire, prevented the accurate calculation of key PSQI components, such as sleep efficiency. As a result, the final analytical sample consisted of 1,478 participants (728 men, 750 women) who provided logically consistent and complete data on the PSQI-J.”

2.1) Also, if participants were contacted in a previous data collection, you need to make this explicit in the main text. It should also be made clear whether all of the data collected was collected during the reporting period (from 18 March 2022 to 22 March 2022) or whether some of the information was obtained from the previous data collection.

Response: Thank you for your helpful comment. All the data used in our study were collected exclusively for our study during the designated survey period (March 18–22, 2022), and no information from previous surveys was used. To make this clear, we have added the following information to the methods section of the main text (page 4, lines 69–74):

“The recruitment period for participation in this study was from March 18 to March 22, 2022. Initially, 1,680 participants were selected for the survey. All participants were adults aged 20–70 years, residing in Japan and registered with Cross Marketing Inc. Some participants had previously participated in other surveys; however, the data used in this study were collected specifically for this project during the designated recruitment period, and no data from previous surveys were included.”

2.2) Has the presence of other medical and/or psychiatric conditions been checked at recruitment? If not, this should be made explicit in the boundaries, as the presence of some of these conditions may affect dream recall.

Response: We acknowledge that psychiatric conditions may affect dream recall and appreciate your bringing this to our attention. In response to your suggestion, we added a sentence to the limitations section clarifying that psychiatric conditions were not specifically assessed during participant recruitment, and therefore, their potential influence on the results cannot be excluded. We also emphasized the need to consider these factors in future research. The following sentence was included (page 18, lines 358–361):

“Moreover, participants in this study were not specifically screened for psychiatric conditions; therefore, the potential influence of such conditions on the results cannot be ruled out. Future studies should consider psychiatric conditions during participant recruitment to better understand their possible impact on dream recall.”

3) The description of how you scaled the Attitudes toward dreams scale should be included in the main text.

Response: Thank you for your helpful comment. In response, we have added a detailed description of the scaling procedure for the Attitudes Toward Dreams Scale in the “Analysis” section (pages 7, lines 156–173). Specifically, we conducted an exploratory factor analysis (EFA) using the principal factor method, removed items with low factor loadings or significant cross-loadings, refined the item pool over five rounds, and performed a confirmatory factor analysis (CFA) to determine the final structure. The final version of the scale comprises two factors with five items each, and each factor’s score was calculated as the mean of the corresponding item scores. We hope this addition clarifies the scaling procedure.

“The results of the Attitudes Toward Dreams Scale questionnaire were first subjected to exploratory factor analysis (EFA) to extract the factor structure. Items with factor loadings below 0.4 or with significant cross-loadings (i.e., loading differences between factors < 0.1) were systematically removed. After the first round of EFA, four items were excluded. In the second round, three additional items were excluded, followed by one item in the third round and two items in the fourth round. Therefore, a total of 10 items were excluded across four rounds of EFA, resulting in 32 items remaining.

After conducting EFA and excluding 10 items, the remaining 32 items were classified into two factors: 22 items in the first factor and 10 items in the second factor. To balance the number of items between the factors, we selected the top 10 items with the highest factor loadings from the first factor. For the second factor, the 10 items retained after EFA were ranked by their factor loadings. Consequently, 10 items were selected from each factor based on the highest loadings, resulting in a 20-item structure. Confirmatory factor analysis (CFA) was then conducted using these 20 items. Subsequently, to identify the model with the best fit, items with the highest factor loadings were selected from both factors, and models with different numbers of items per factor (e.g., eight items, seven items, and five items) were tested. The model with five items per factor demonstrated the highest model fit indices. Therefore, the final version of the Attitudes Toward Dreams Scale consisted of 10 items, with five items representing each factor.”

Result

4) Age-related ds should also be added to Table 1. Also, Actual% 2020 refers to what? You should describe this in the heading.

Response: Thank you for your valuable feedback. To address your comment, we have revised Table 1. We have added the standard deviation for age and clarified the meaning of “Actual% (2020)” in the table’s footnote, indicating that it refers to the age and sex distribution of the Japanese population based on the 2020 national census data (page 8, line 192– page 9, line 195).

5) A table with descriptive analyses of any other measures you collected in the survey should be included.

Response: To address this valuable suggestion, we have created a new table (Table 2) that presents the descriptive statistics of the additional measures collected in the survey. Table 2 has been added to the revised manuscript on page 9, line 200 – page 10, line 205.

We have also added a new section titled “Questionnaire scoring and cutoff criteria” to explain the scoring and cutoff values for each item in the survey (pages 5–6, lines 118–133). Furthermore, regarding the STOP-Bang, we have clarified in the “Survey items” section that only four of the eight questions were used for the purpose of our study (page 5, lines 104–107).

“Questionnaire scoring and cutoff criteria

In this study, we used a variety of psychological and sleep-related questionnaires, each with specific scoring characteristics and interpretive guidelines. The Japanese version of the RBD scale (RBDQ-J) serves as a screening tool for REM sleep behavior disorder, with scores of 5 or higher suggesting probable RBD. For the STOP-Bang questionnaire, only the four STOP items (snoring, tiredness, observed apnea, and high blood pressure) were included in this study; therefore, no established cutoff score is applicable. The RLS scale is used to assess symptom frequency, with higher scores indicating more frequent symptoms. The Athens Insomnia Scale is used to screen for insomnia, with scores of 6 or higher suggesting possible insomnia. The PSQI-J is used to measure subjective sleep quality, with a global score of 6 or higher indicating poor sleep quality. The Japanese version of the Metamemory in Adulthood Questionnaire is used to evaluate beliefs and awareness regarding memory functioning, and the NEO-FFI is used to assess five personality traits. In both instruments, higher scores reflect stronger trait expression. The Dream Propensity Scale captures dream-related tendencies; however, no defined cutoff score is provided. The SF-8 is used to measure health-related quality of life using norm-based scoring (mean = 50, standard deviation = 10), with higher scores indicating better perceived health status.” (pages 5 line 118– page 6 line 133)

“…questionnaire for the evaluation of obstructive sleep apnea (STOP-Bang) [15] (only the four STOP items: snoring, tiredness, observed apnea, and high blood pressure were included; the BANG items—BMI, age, neck circumference, and gender— were not included and were therefore not obtained from the participants);” (page 5, lines 104 – 107)

6) Table 4 still shows several measures such as blood pressure, BMI, age, neck circumference, gender, but not the data from these measures?

Response: Thank you for highlighting this. We have added a new Table 2 to the manuscript; therefore, the original Table 4 has been renumbered as Table 5 (page 13, line 244 – page 14, line 249).

In this study, we did not administer all items of the STOP-Bang questionnaire; only the four STOP items—snoring, tiredness, observed apnea, and high blood pressure—were included in the survey. Two of the BANG items—BMI and neck circumference—were not part of the questionnaire and were not obtained from the participants. However, age and gender (sex) were collected as part of the basic demographic information and are included in analyses elsewhere in the paper (e.g., comparisons across different age groups and sexes in other tables). The inclusion of these items in the original Table 4 in the context of the STOP-Bang was an error, which we have corrected. We sincerely apologize for this oversight.

---

## [Decision Letter · Decision Letter 2]

Creating a Japanese version of the Attitudes Toward Dreams Scale: Attitude toward dreams may predict sleep disorders

PONE-D-24-51943R2

Dear Dr. Okuyama,

We’re pleased to inform you that your manuscript has been judged scientifically suitable for publication and will be formally accepted for publication once it meets all outstanding technical requirements.

Kind regards,

Serena Scarpelli

Academic Editor

PLOS ONE

Additional Editor Comments (optional):

Reviewers' comments:

Reviewer's Responses to Questions

**Comments to the Author**

Reviewer #3: All comments have been addressed

2. Is the manuscript technically sound, and do the data support the conclusions?

Reviewer #3: Partly

3. Has the statistical analysis been performed appropriately and rigorously?

Reviewer #3: Yes

4. Have the authors made all data underlying the findings in their manuscript fully available?

Reviewer #3: Yes

5. Is the manuscript presented in an intelligible fashion and written in standard English?

Reviewer #3: Yes

Reviewer #3: The manuscript has been improved with revisions. However, one issue remains:

STOP-BANG questionnaire: Since only part of it was administered, it is necessary to explicitly justify this methodological choice, explain its consistency with the literature, and include the relevant reference.

If a satisfactory justification cannot be provided, I would recommend removing this variable from the analysis and noting in the limitations section the absence of information on nocturnal sleep apnea."

**Do you want your identity to be public for this peer review?** For information about this choice, including consent withdrawal, please see our Privacy Policy

Reviewer #3: No

---

## [Editor Report · Acceptance letter]

PONE-D-24-51943R2

PLOS ONE

Dear Dr. Okuyama,

I'm pleased to inform you that your manuscript has been deemed suitable for publication in PLOS ONE. Congratulations! Your manuscript is now being handed over to our production team.

Kind regards,

on behalf of

Dr. Serena Scarpelli

Academic Editor

PLOS ONE